# Revealing defective interfaces in perovskite solar cells from highly sensitive sub-bandgap photocurrent spectroscopy using optical cavities

Bas T. van Gorkom[1], Tom P. A. van der Pol [1], Kunal Datta [1], Martijn M. Wienk[1] & René A. J. Janssen [1,2✉]

Defects in perovskite solar cells are known to affect the performance, but their precise nature, location, and role remain to be firmly established. Here, we present highly sensitive measurements of the sub-bandgap photocurrent to investigate defect states in perovskite solar cells. At least two defect states can be identified in p-i-n perovskite solar cells that employ a polytriarylamine hole transport layer and a fullerene electron transport layer. By comparing devices with opaque and semi-transparent back contacts, we demonstrate the large effect of optical interference on the magnitude and peak position in the sub-bandgap external quantum efficiency (EQE) in perovskite solar cells. Optical simulations reveal that defects localized near the interfaces are responsible for the measured photocurrents. Using optical spacers of different lengths and a mirror on top of a semi-transparent device, allows for the precise manipulation of the optical interference. By comparing experimental and simulated EQE spectra, we show that sub-bandgap defects in p-i-n devices are located near the perovskite-fullerene interface.

---

[1] Molecular Materials and Nanosystems, Institute for Complex Molecular Systems, Eindhoven University of Technology, P.O. Box 513, 5600 MB Eindhoven, The Netherlands. [2] Dutch Institute for Fundamental Energy Research, De Zaale 20, 5612 AJ Eindhoven, The Netherlands. ✉email: r.a.j.janssen@tue.nl

Defects states have been the subject of numerous studies since the rise of perovskite photovoltaics[1–4], but much remains unclear about their exact energetic nature and impact on device performance. Studies have shown the presence of significant defect densities[5,6], yet this does not seem to impact device performance significantly given the high efficiencies that have been achieved[7]. Extensive compositional engineering has been done to achieve these efficiencies[8–10], but still the effect on the structural disorder and the formation of energetic states in the bandgap is poorly understood[11]. Defects have also been shown to form upon illumination[12,13], negatively impacting stability and performance under operating conditions. Better understanding the origin and properties of defects in hybrid organic–inorganic perovskite photovoltaics is therefore vital to not only reduce nonradiative losses but to also provide solutions for stability issues, ion migration, and halide segregation[14,15]. Given their nature, the detection of defect states requires highly sensitive characterization techniques. Measurements of sub-bandgap photocurrents as a result of defect states present in the bandgap have been used before to study defect properties in silicon-based photovoltaics using constant photocurrent methods[16,17]. In addition, Fourier-transform photocurrent spectroscopy has been used to study sub-bandgap signals in numerous photovoltaic materials[18,19], including perovskites[20]. More recently, variations on these techniques have been used to study the sub-bandgap external quantum efficiency (EQE) to gain more information on the energetic distribution of the defects found in various photovoltaic materials[21–23]. Here, we use a similar, highly sensitive measurement of the EQE to study the sub-bandgap signals in p–i–n perovskite solar cells and demonstrate the significant impact of optical interference on these signals. Not accounting for these effects can lead to misinterpretation of the sub-bandgap signals and result in invalid conclusions on the defect energies. By combining the sub-bandgap photocurrent measurements with optical simulations for different back contacts, we show the extent of the spectral distortion caused by optical interference. We further use optical spacers of different path lengths to systematically vary the optical interference. From the changes in the sub-bandgap EQE with the size of the optical cavity, we infer that at least one of the defects in p–i–n perovskite solar cells is located at the i–n interface.

Generation of photocurrent typically occurs by absorption of a photon with an energy equal or larger than the bandgap, leading to excitation of an electron from the valence band to the conduction band, also referred to as a band-to-band transition. In a perfect, defect-free, direct bandgap semiconductor these are the only transitions that occur. In practice, there are always inhomogeneities in the density of states (DOS) near the band edges that lead to an exponential tail in the photocurrent spectrum near the band edge. The absorption coefficient in that sub-bandgap spectral range is well described by Urbach's rule[24]:

$$\alpha(E) = \alpha_0 \exp\left(\frac{E - E_g}{E_u}\right) \qquad (1)$$

with $\alpha_0$ a material constant, $E$ the photon energy, $E_g$ the bandgap, and $E_u$ the Urbach energy. The Urbach energy is a parameter for the energetic disorder in the material and for perovskites typically lies around 15–20 meV[23,25], surprisingly low for a polycrystalline material processed at low temperatures. When additional photocurrent is measured at photon energies even lower than those of the Urbach tail, electronic transitions involving states in the bandgap are involved. For perovskites, these are attributed to the presence of defect states, either point defects (e.g., vacancies, interstitials, and anti-sites) and line defects in the perovskite lattice, or macroscopic defects like grain boundaries[26,27].

Assuming the defect states have a Gaussian distributed DOS and assuming only band-to-defect or defect-to-band transitions, their contribution to the EQE will follow:

$$\text{Defect EQE}(E) = \kappa \frac{G_d}{2} \left( 1 + \text{erf}\left(\frac{E - E_d}{\sigma_d \sqrt{2}}\right) \right) \qquad (2)$$

with $G_d$ the defect density, $E_d$ the defect energy, $\sigma_d$ the standard deviation of the Gaussian defect DOS, and $\kappa$ a proportionality constant that involves, amongst others, the absorption coefficient and charge collection efficiency. This equation has been used to fit sub-bandgap states for numerous semiconductors[18,22,23,28,29]. Any sub-bandgap EQE spectrum can then be fitted by a combination of an Urbach tail described by Eq. (1) and a distinct number of defect contributions described by Eq. (2). From such fits, information can be obtained on the energy of the defect states and, via the intensity of the measured signal, on the defect density.

## Results

To investigate the presence of defect states and obtain information on their energetic properties and location in the device, we examined a series of p–i–n solar cells consisting of a double-cation FA$_{0.67}$MA$_{0.33}$PbI$_{2.85}$Br$_{0.15}$ perovskite (FA is formamidinium, MA is methylammonium) sandwiched between a poly[bis(4-phenyl)(2,4,6-trimethylphenyl)amine] (PTAA) hole transport layer deposited on a glass substrate covered with an indium tin oxide (ITO) front electrode and a [6,6]-phenyl-C$_{61}$-butyric acid methyl ester (PCBM) electron transport layer covered with aluminum-doped zinc oxide (AZO). The perovskite films were deposited via a two-step procedure involving deposition of PbI$_2$ in the first step and of organic halides (methylammonium iodide (MAI), methylammonium bromide (MABr), and formamidinium iodide (FAI)) in the second step, followed by thermal annealing. X-ray diffraction (XRD) and scanning electron microscopy (SEM) confirm the identity and polycrystalline nature of the perovskite films (Supplementary Fig. 1). Current density–voltage (J–V) curves and EQEs of opaque and semi-transparent cells with Al and ITO back contacts are shown in Supplementary Fig. 2. The cells provide power conversion efficiencies of 17.9% (for Al) and 14.7% (for ITO) in reverse scans with small hysteresis.

A typical sub-bandgap EQE spectrum for these cells is shown in Fig. 1a. By minimizing noise (see "Methods" for details), the EQE can be measured over nine orders of magnitude. The $E_u$ is about 14 meV, consistent with reported values for perovskites[20,23]. Below the exponential tail, two broad features are visible, which would point towards the contribution of at least two distinct defect states, similar to what has been reported for equivalent measurements[23]. However, when moving from low to high photon energies, there are places in the spectrum where the EQE signal drops, e.g., the signal at 1.0 eV is lower than at 0.9 eV. If the contribution of defects to the EQE is a result of a sum of functions as described by Eq. (2), this decrease in measured photocurrent would imply a negative DOS. One supposed explanation for this illogical observation is the impact of optical interference on the measured EQE in the sub-bandgap region. The effect of optical interference on these types of signals has been described for amorphous-Si[18] and organic photovoltaic devices[30,31], showing a distortion of the (sub-bandgap) EQE signals as a result of interference of the light, varying with active layer thickness. While these effects are not necessarily surprising because absorption below the bandgap is negligible and therefore interference of the sub-bandgap probe light is likely, similar effects have thus far not been described for perovskite devices. In order to gain a better understanding of the magnitude of any

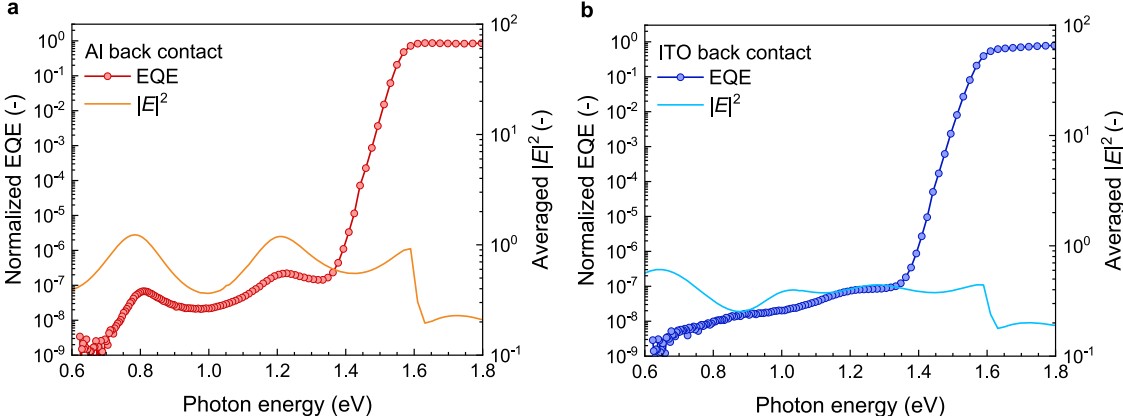

**Fig. 1 Experimental EQE and averaged simulated $|E|^2$ of a p–i–n perovskite solar cell with a glass/ITO/PTAA/FA$_{0.67}$MA$_{0.33}$PbI$_{2.85}$Br$_{0.15}$/PCBM/AZO/ back electrode configuration. a** For an Al back electrode. **b** For an ITO back electrode. The variation in $|E|^2$ is much larger when using Al as the back electrode than for ITO due to the higher reflectivity of Al. The modeled interference peaks line up well with peaks in the measured EQE. $|E|^2$ has been averaged over the full thickness of the perovskite layer.

optical interference effects on the measured sub-bandgap EQE signals, the normalized modulus squared of the optical electric field $|E|^2$ in the device stack is modeled using a script based on the transfer matrix method[30,32]. The wavelength-dependent refractive index ($n$) and extinction coefficient ($k$) of the layers used in the optical modeling of the solar cells are shown in Supplementary Fig. 3. Figure 1a shows $|E|^2$ as a function of photon energy averaged over the thickness of the perovskite film for a glass/ITO/ PTAA/FA$_{0.67}$MA$_{0.33}$PbI$_{2.85}$Br$_{0.15}$/PCBM/AZO/Al solar cell. Below the bandgap the variation in $|E|^2$ ranges between 0.3 and 1.35, indicating the extent to which sub-bandgap defects in the bulk contributing to the EQE will be distorted due to this variation. Two peaks are observed below the bandgap at around 0.8 and 1.2 eV. These peaks coincide with the peaks in the measured photocurrent spectrum. This indicates that optical interference indeed affects sub-bandgap photocurrents in perovskite devices. The main cause for $|E|^2$ to vary strongly is the negligible absorption in the sub-bandgap region combined with the high reflectivity of the metal back electrode. The distortion caused by constructive and destructive optical interference affects the intensity of the defect-related signals and complicates fitting the measured spectra and deriving energetic parameters of the defect states.

For a better understanding of the extent to which optical interference affects sub-bandgap photocurrent signals in perovskite solar cells, a device configuration where the interference is minimized is required. One way to reduce the interference-related variation in $|E|^2$ is to minimize the reflection at the back contact as much as possible by replacing the reflecting metal contact with a transparent electrode, e.g., ITO. Replacing the aluminum back contact with ITO indeed reduces the variation in $|E|^2$, which now ranges from 0.35 to 0.6 over the entire sub-bandgap region (Fig. 1b). The reduced variation in $|E|^2$ with a transparent back electrode is reflected in the sub-bandgap EQE spectrum (Fig. 1b), where the EQE is significantly more uniform for photon energies below 1.4 eV. The experimental sub-bandgap EQE spectrum is now more consistent with the expected peak shapes as described by Eq. (2). The effect of illumination direction on the $|E|^2$ of the semitransparent device is small (Supplementary Fig. 4).

To investigate and understand the effect of interference in more detail, the EQE was measured varying systematically the length of the optical cavity by adjusting the device stack. Optical interference mainly depends on the wavelength-dependent refractive indices and extinction coefficients, and on thicknesses of the various layers in the optical stack. Any changes to the layer

thickness will therefore impact the optical electric field distribution through the device stack. As the perovskite layer is the thickest, it seems obvious to vary the thickness of this layer as has been done for other semiconductors to show changes in the sub-bandgap EQE[18,30]. However, due to the two-step perovskite deposition process used here, a constant quality of the perovskite layer in terms of optical properties and defect densities cannot be guaranteed. Changes in grain size, e.g., can impact charge transport properties, defect densities, and optical properties[33–35]. Likewise, varying the thickness of the transport layers can impose problems with charge carrier collection impeding the photovoltaic performance of the cell, which might, in turn, affect the sub-bandgap EQE[36]. To decouple the effects of optical interference on the sub-bandgap EQE from changes due to variations in the performance of the cell, an optical cavity that does not impact the electronic properties of the cell is desirable. Here, this is realized by placing an optical spacer layer behind the ITO back electrode, followed by a silver layer acting as a mirror. In this way, the sub-bandgap EQE of a semitransparent device can be measured before and after adding this spacer-mirror stack. Moreover, the thickness of the optical spacer layer can be varied at will to shift the optical electric field in the active layer. As all layers of the underlying perovskite cell now remain unchanged, any distortions observed in the sub-bandgap EQE are solely caused by differences in the optical electric field. The optical spacer layer should be nonconductive to not interfere with cell performance and transparent in the NIR region to prevent parasitic absorption of the probe light. It should be noted that commercial ITO typically shows an absorption in the near-infrared (NIR) region. However, for the sputtered ITO back contact excellent NIR transmission can be achieved, while obtaining sufficiently low sheet resistances (Supplementary Fig. 5). To allow for precise control over the thickness, magnesium fluoride (MgF$_2$), deposited via thermal vapor deposition, was used as an optical spacer. The changes in optical interference as a result of the spacer-mirror layers can be simulated using optical modeling. Then, by comparing the modeled and experimental data, the extent to which the optical interference affects the sub-bandgap EQE can be quantified.

The sub-bandgap EQE of four identical semitransparent p–i–n devices was measured, after which four different thicknesses of MgF$_2$ (100, 150, 200, and 300 nm) and a layer of Ag (100 nm) were added on top of the cells. The sub-bandgap EQE was then measured once more (Fig. 2). In their semitransparent state, the cells show identical sub-bandgap EQEs (Fig. 2a), which confirms

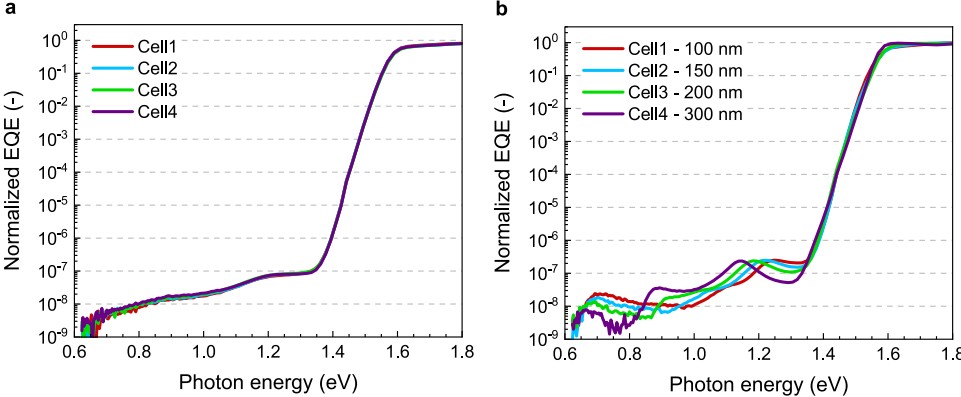

**Fig. 2 Sub-bandgap EQE of four p–i–n perovskite solar cells using an ITO back electrode. a** Without optical spacer or mirror. **b** With an MgF$_2$ optical spacer and a Ag mirror. The thickness of the MgF$_2$ layers was 100, 150, 200, and 300 nm as indicated in the legend.

that there is very little variation between these nominally identical cells. Upon addition of the spacer-mirror layers, the sub-bandgap EQE changes drastically for all cells (Fig. 2b). Not only when comparing the different spacer thicknesses but also when comparing the spectra with their semitransparent equivalents. It is emphasized that measurements with and without spacer were performed on the same cell and only the optics in the layer stack have changed. With the Ag mirror, a clear modulating interference pattern emerges in the sub-bandgap region of the EQE spectrum. The peaks shift to lower energies with increasing thickness of the spacer layer because the optical cavity becomes longer. This is corroborated by optical modeling of the layer stack for different thicknesses of the MgF$_2$ layers (Supplementary Fig. 6a). Apart from a shift in their positions, also the intensities of the peaks change. Notably, at certain energies, the intensity differences between samples can be as much as half an order of magnitude, depending on the spacer thickness. For example, the sub-bandgap EQE at 1.15 eV increases from $5.5 \times 10^{-8}$ for a spacer thickness of 100 nm to $2.4 \times 10^{-7}$ for a 300 nm spacer. These large differences are impossible to explain when considering the typical values for $|E|^2$ averaged over the entire active layer (Fig. 1a). In addition, there are energies at which the devices with the spacer-mirror layers have a significantly lower signal compared to their semitransparent state, e.g., at 1.3 eV for 100 nm MgF$_2$. These can also not be explained by the modeled results because $|E|^2$ for the spacer-mirror devices is equal or higher than for the semitransparent cells over the entire energy range (Fig. 1). The large distortions that result from optical interference complicate obtaining accurate defect energies from the sub-bandgap EQE, and omitting these effects will lead to gross inaccuracies, as has been described for organic solar cells by Kaiser et al.[31]. and for amorphous silicon solar cells by Melskens et al.[18]. Clearly, a semitransparent device can give a more accurate estimate because the variations in the optical electric field are less compared to cells with a reflective back contact. But even in semitransparent cells, interference is not negligible and should still be taken into consideration.

To explain the substantial differences that are measured when adding the spacer-mirror stack, the spatial distribution of the optical electric field within the device stack for different configurations is investigated in more detail (Fig. 3). Constructive and destructive interference gives rise to minima and maxima in the optical electric field $|E|^2$ at certain positions across the device stack and for given photon energies several maxima and minima fit within the width of the perovskite layer. Therefore, averaging $|E|^2$ over the full thickness of the perovskite layer moderates the interference and predicts differences that are much smaller than those measured experimentally.

To align the measured and modeled data, a situation can be considered where parts of the active layer contribute significantly more to the defect-related EQE of the photocurrent than others. For perovskites, the interfaces between the active layer and the charge transport layers have a well-known reputation for being defect-rich in nature because the perovskite crystallites terminate here and are therefore a source of under coordinated ions that can act as trap states[37,38]. While direct observation and characterization remains difficult, numerous surface passivation strategies have been employed over the years to passivate interfacial trap states and reduce nonradiative recombination, resulting in observed increases of the open-circuit voltage[39–42]. Alternatively, the defects are distributed homogenously, but only charge carriers generated close to the interface are extracted, generating a photocurrent.

To simulate the contribution of only the interfaces between the perovskite absorber and the charge transport layers to the sub-bandgap photocurrents, the effect of $|E^2|$ can be weighted using a half-normal distribution centered at the interfaces of the perovskite layer with PTAA and/or PCBM transport layers and extending into the perovskite layer with a standard deviation of 30 nm (Supplementary Fig. 7). The half-normal distribution is used to account for the defect density that is expected to gradually decrease when moving from the interface to the perovskite bulk. It also mimics the effects surface roughness and small thickness variations across the device stack have on the optical electric field. Weighting $|E|^2$ results in interference patterns with much higher maxima and lower minima (Supplementary Fig. 6b, c), which is in accordance with the large modulations measured and with the large differences between cells with different spacer thicknesses. This supports the hypothesis that defect states present at one or both interfaces are responsible for the measured sub-bandgap photocurrents. It is then the question of whether the defect-related EQE can be attributed to one particular interface, or that defects at both interfaces are contributing. When looking at the standard device layout with an aluminum back contact, this question is strenuous to answer conclusively, because the interference patterns at the front PTAA interface and back PCBM are virtually interchangeable. Both show two peaks at roughly the same energies, varying only slightly in intensity (Supplementary Fig. 6b, c). However, the use of differently sized optical cavities allows us to distinguish between contributions of different parts of the perovskite layer. Since the thickness of the MgF$_2$ can be varied at will, situations can be created where the front and back interfaces have different interference patterns. Using optical modeling of the electric field for different MgF$_2$ thicknesses, the expected sub-bandgap EQE can be calculated for selected contributions at different positions in the perovskite layer. By

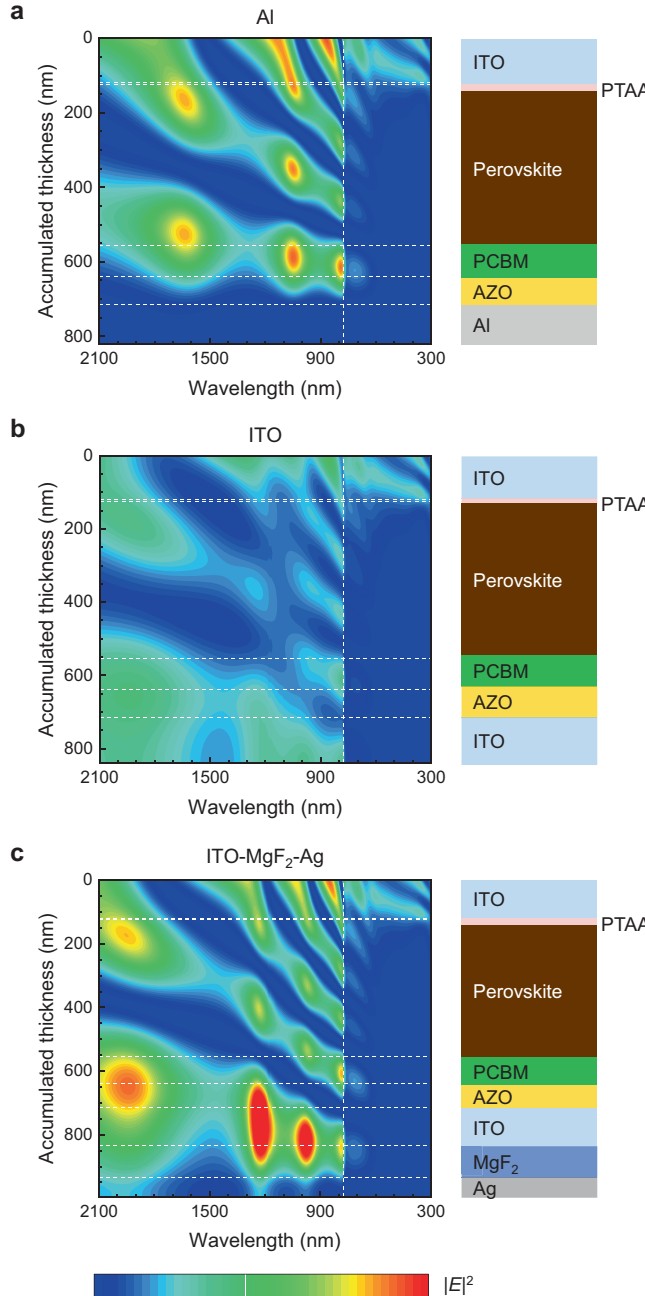

**Fig. 3 Modeled spatial distribution of $|E|^2$ in glass/ITO/PTAA/ FA$_{0.67}$MA$_{0.33}$PbI$_{2.85}$Br$_{0.15}$/PCBM/AZO/back electrode p–i–n devices. a** With a metal (Al) back electrode. **b** With a transparent (ITO) back electrode. **c** With a transparent (ITO) back electrode, an additional optical spacer (MgF$_2$), and a back reflector (Ag). The thickness of the MgF$_2$ layer used in the simulation shown was 100 nm.

comparing these modeled results with the measured EQE, we can more conclusively determine which interface is dominating the defect-related EQE, and thus, which interface is defect-rich in nature. Based on the data measured for the semitransparent devices (Fig. 2), two defect functions (Eq. (2)) at photon energies $E_d = 0.72$ and $E_d = 1.09$ eV, with $\sigma_d = 0.04$ eV were used to mimic the expected EQE (see Supplementary Fig. 8 and Supplementary Note 1 for details). The sum of these functions is then multiplied by $|E|^2$, which is averaged over either the entire

perovskite layer, or only at the PTAA–perovskite or perovskite–PCBM interface for every different thickness of MgF$_2$. The resultant modeled EQEs are compared to the experimental data in Fig. 4.

## Discussion

The spectra modeled by averaging $|E|^2$ over the full perovskite layer display modulations that are significantly smaller than the measured data. Modeling the sub-bandgap EQE at the interfaces instead of the perovskite bulk yields spectra with higher variation in signal over a relatively narrow energy range, consistent with the measured data. Modeling the EQE as arising from the PTAA–perovskite interface only predicts a distinct minimum at 1.03 to 1.12 eV for an MgF$_2$ layer varying from 100 to 300 nm (Fig. 4), due to significant destructive interference. This sharp minimum is, however, absent in all experimental EQE spectra. On the other hand, when modeling the sub-bandgap photo-current generation to occur only at the perovskite–PCBM inter-face, the predictions closely resemble the measured EQE. For an MgF$_2$ thickness of 100 nm, a local minimum at around 1.0 eV is seen in both the measured and calculated data. In addition, when increasing the MgF$_2$ thickness, the signal increases and a local maximum is seen for 300 nm of MgF$_2$ at an energy of around 0.88 eV. The distinction between the different scenarios (con-tribution from the interface of the perovskite layer with either PCBM or PTAA) is quite clear below 1.1 eV, where only the low photon energy defect absorbs strongly, indicating that this defect is located at the perovskite–PCBM interface.

The calculated EQE at higher photon energies is relatively similar for both scenarios, with a local minimum at the bottom of the Urbach tail at around 1.35 eV that deepens with increasing spacer thickness. Therefore, it is less obvious to identify at which of the two interfaces the higher energy defect is located. We also considered the situation in which the two defects are located at opposite interfaces. Supplementary Fig. 9 shows simulated EQE spectra when one defect is placed at the PTAA–perovskite interface and the other at the perovskite–PCBM interface. Comparison of these spectra with the experimental EQE also show much better resemblance when the low photon energy defect is assumed to be located near the perovskite–PCBM interface, substantiating that the low photon energy defect is located at this interface. For the high photon energy defect close to the Urbach tail, contributions from the PTAA–perovskite interface cannot be ruled out. These results underline the need for suitable surface passivation treatments of the perovskite layer in p–i–n solar cells after fabrication.

In principle, the charge transport layers can also contribute to the sub-bandgap EQE. The simulated EQE spectra in which the defects are assumed to be located in the PTAA and PCBM charge transport layers (Supplementary Fig. 10) are similar to those modeled for defects at the PTAA–perovskite and perovskite–PCBM interfaces, respectively (Fig. 4). Defects located in the PTAA layer are predicted to result in very strong destructive interference for photon energies between 0.9 and 1.1 eV and between 1.25 and 1.35 eV (Supplementary Fig. 10), which is not observed experimentally (Fig. 4), but a contribution of the PCBM electron transport layer cannot be excluded.

The results demonstrate the significant influence of optical interference on the sub-bandgap EQE of perovskite solar cells. The sub-bandgap EQE is distorted by strong variations in the optical electric field as a result of negligible absorption in the perovskite layer and a high reflection of traditional metal back contacts. These effects lead to changes in the measured sub-bandgap EQE as large as one order of magnitude, complicating any fitting of the defect-related signals and subsequent extraction

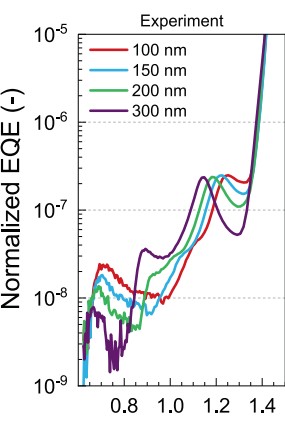
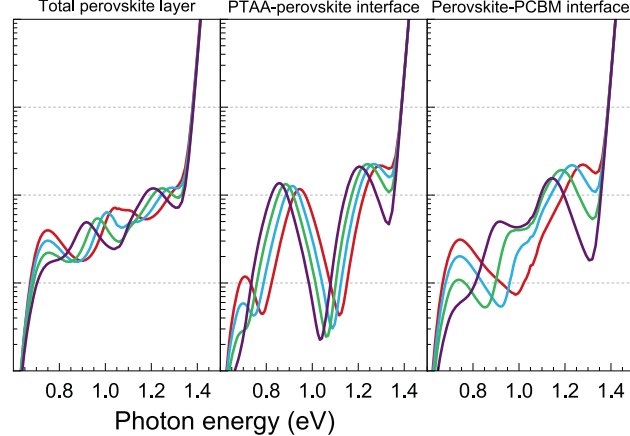

**Fig. 4 Experimental and modeled EQE spectra of perovskite solar cells with a transparent back contact and a spacer mirror on top.** The simulated EQEs result from two defect functions (Eq. (2)) at photon energies $E_d = 0.72$ and $E_d = 1.09$ eV, with $\sigma_d = 0.04$ eV, multiplied by $|E|^2$ after averaging over the entire perovskite layer, or over a half-normal distribution with a standard deviation of 30 nm, centered at the PTAA–perovskite or at the perovskite–PCBM interface. An Urbach tail has been added to enable comparison with the experimental data. The legend gives the thickness of the $MgF_2$ optical spacer positioned between the transparent ITO back electrode and the Ag mirror (see Fig. 3c).

of any energetic parameters for these defects. We emphasize that these findings are not only relevant for sub-bandgap photocurrent measurements, but any characterization method using sub-bandgap probe light. By using a transparent back electrode, these effects can be reduced, and a more reliable defect-related EQE can be obtained. In addition, by creating an optical cavity using $MgF_2$ layers of different thicknesses and a Ag mirror, the effect of optical effects can be decoupled from the photovoltaic performance. By combining this method with optical modeling, we show that for PTAA/$FA_{0.67}MA_{0.33}PbI_{2.85}Br_{0.15}$/PCBM p–i–n devices states near the perovskite–PCBM interface dominate the lower energy part of the photocurrent spectrum and that defects are thus located near this interface. These findings underline the need for appropriate consideration of optical interference effects when interpreting any sub-bandgap spectrum, imperatively so for perovskite devices given the strongly localized effect due to the defect-rich nature of the interfaces. Also, it highlights the need for suitable surface passivation treatments to reduce the defect density and minimize nonradiative losses. With this method, we have been able to identify the location and the energy of the defects, but not yet their exact nature. For optically excited sub-bandgap defects that generate photocurrent in organic photovoltaic devices an optical release mechanism involving mid-bandgap states has recently been proposed by Zarrabi et al.[21] A similar mechanism may explain defect-mediated charge generation in perovskites, although the sub-bandgap photon energies of 0.72 and 1.09 eV that generate charges are not exactly mid-bandgap ($E_g = 1.574$ eV). Alternatively, a direct interfacial charge transfer from an occupied trap into the lowest unoccupied molecular orbital of PCBM can also contribute.

## Methods
**Solution preparation**. Lead iodide ($PbI_2$, 99.99%, trace metal basis, TCI Chemicals) (553 mg) was dissolved in a mixture of dimethylformamide (Sigma-Aldrich, anhydrous, 99.8%) (0.876 mL) and dimethyl sulfoxide (Sigma-Aldrich, 99.9%) (0.0864 mL) and stirred overnight at 60 °C inside a nitrogen-filled glovebox. MAI (14.27 mg), 7.57 mg, and FAI (53.97 mg), obtained from Greatcell Solar Materials, were dissolved in 2-propanol (Sigma-Aldrich, 99.5%) (1 mL) and stirred overnight at 60 °C in a nitrogen-filled glovebox. PTAA ($M_w = 14.5$ kg mol$^{-1}$, EM Index Co. Ltd) (3 mg) was dissolved in toluene (Sigma-Aldrich, anhydrous) (1 mL) and stirred overnight at 60 °C in a nitrogen-filled glovebox. PCBM (99%, Solenne BV) (20 mg) was dissolved in a 1:1 (v/v) mixture of chlorobenzene and chloroform (1 mL) and stirred overnight under ambient conditions. A dispersion of AZO nanoparticles (Avantama, 2.5 wt% in alcohols) was stirred overnight in a nitrogen-filled glovebox at ambient temperature.

**Film characterization**. XRD was measured with a Bruker 2D Phaser using a Cu Kα ($\lambda = 1.5405$ Å) X-ray source. The increment step size was 0.02° between 10° and 40° and an acquisition time of 1.5 s was used at each increment. The surface morphology of the perovskite layer was characterized by scanning electron microscopy (Thermo Fisher Scientific, Quanta 3D FEG) using a 10 kV electron beam and a secondary electron detector.

**Device fabrication**. Patterned ITO substrates (Naranjo Substrates) were cleaned by sonication in acetone, scrubbing and sonication in a solution of sodium dodecyl sulfate (Acros, 99%) in water, rinsed with deionized water, and sonicated in 2-propanol. The substrates were dried and treated in an ultraviolet (UV)–ozone oven for 30 min, shortly before use. The substrates were then transferred to a nitrogen-filled glovebox for spin coating of the charge transport and perovskite layers. The PTAA solution was spin coated at 5800 r.p.m. for 30 s, followed by annealing at 100 °C for 10 min. The substrate was then allowed to cool to room temperature. The perovskite layer was spin coated using a two-step process where the $PbI_2$ solution was spin-coated at 3000 r.p.m. (at an acceleration of 2000 r.p.m. s$^{-1}$) for 30 s, after which the organic halide solution was cast onto the wet $PbI_2$ film at 3300 r.p.m. (at an acceleration of 20,000 r.p.m. s$^{-1}$) for 30 s. The perovskite film was thereafter annealed at 100 °C for 30 min, yielding a dark, shiny film. PCBM was spin coated at 1000 r.p.m. for 60 s, followed by an annealing step at 100 °C for 30 min. AZO was spin coated twice at 2000 r.p.m. for 60 s, followed by annealing at 75 °C for 1 min and 5 min after the first and second spin coating step, respectively. Aluminum, $MgF_2$, and silver were all thermally evaporated onto the samples under a high vacuum (~$3 \times 10^{-7}$ mbar). The ITO back electrodes were made via radio frequency magnetron sputtering. The overlap between the ITO front and Al, or ITO, back contact (0.09 cm$^2$) determined the active area of the solar cells.

**EQE measurements**. To measure EQE, the cells were contacted in a nitrogen-filled container. A 50 W tungsten–halogen lamp was used as the light source. The light was chopped at 158 Hz before passing into a monochromator (Oriel, Cornerstone 130). A reference silicon detector was used to calibrate the current from the cell, which was fed into a current preamplifier (Stanford Research, SR 570). The resulting voltage was measured using a lock-in amplifier (Stanford Research, SR830). A green (Thorlabs, M530L3) LED was used as a light bias to generate approximately 1-Sun equivalent illumination intensity. Integration of the EQE with the AM1.5 G spectrum afforded estimates of $J_{sc}$ that were within 4% of the values measured with the simulated solar light.

**Sensitive photocurrent spectroscopy**. To measure EQE in the sub-bandgap region, an Oriel 3502 light chopper, Cornerstone 260 monochromator (CS260-USB-3-MC-A), a Stanford Research SR 570 low-noise current preamplifier, a Stanford Research SR830 lock-in amplifier, and a 250 W tungsten–halogen lamp were used. The light was chopped at a frequency of 330 Hz. A series of long-pass filters (OD ≥ 5) with increasing cut-on wavelengths was placed between the lamp and monochromator to remove stray light during the measurement. The monochromatic light is then passed through a concave cylindrical lens, to focus the light and increase the intensity on the active area of the solar cell. The solar cell was kept in an electrically insulated nitrogen-filled container. The cell is contacted using spring-loaded gold contacts and the container is contacted with a LEMO connector. The current generated by the solar cell is fed into the preamplifier via a

triaxial cable, which is kept at a distance from other cabling to minimize the spurious signals due to induction. Above the bandgap, a sensitivity of $200\,\mu A\,V^{-1}$ was used for the preamplifier and this was increased to $200\,nA\,V^{-1}$ to measure signals below the bandgap. The lock-in amplifier was set in float mode to reduce background noise and a time constant of 1 s and a settling time of 15 s were used. Calibrated Si and InGaAs photodiodes were used to determine incident light intensity.

**Device characterization.** Optical simulations were carried out based on the transfer matrix method using an in-house adapted version of the code provided by Burkhard et al. [32]. $J$–$V$ characteristics of the solar cells were measured with a Keithley 2400 SMU in a dry and oxygen-free nitrogen atmosphere (<1 p.p.m. $O_2$ and <1 p.p.m. $H_2O$). A tungsten–halogen lamp, filtered by a Schott GG385 UV filter and a Hoya LB120 daylight filter, was used to simulate the air mass 1.5 globally diffuse (AM1.5 G) solar spectrum at $100\,mW\,cm^{-2}$, calibrated by Si photodiode. No preconditioning of the cells was used before characterization. A shadow mask with $0.0676\,cm^2$ was used to define the cell area. $J$–$V$ scans involved sweeping the applied voltage (with no prebiasing) from $+1.5$ to $-0.5\,V$ for a reverse scan or from $-0.5$ to $+1.5\,V$ for a forward scan at a rate of $0.25\,V\,s^{-1}$.

**Reporting summary.** Further information on research design is available in the Nature Research Reporting Summary linked to this article.

## Data availability

All relevant data in this study are available from the corresponding author upon request. Source data are provided with this paper.

## Code availability

The code used for optical simulations is available from the corresponding author upon request.

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

## Acknowledgements

This work is part of the Advanced Research Center for Chemical Building Blocks, ARC CBBC, which is cofounded and cofinanced by the Netherlands Organisation for Scientific Research (NWO) and the Netherlands Ministry of Economic Affairs (project 2016.03.Tue) (B.T.v.G., R.A.J.J.). We further acknowledge funding from the Ministry of Education, Culture, and Science (Gravity program 024.001.035) (T.P.A.v.d.P., R.A.J.J.) and from NWO via and the Joint Solar Programme III (project 680.91.011) (K.D., R.A.J.J.) and an NWO Spinoza grant (M.M.W, R.A.J.J.).

## Author contributions

B.T.v.G. performed the photocurrent experiments. T.P.A.v.d.P. made the optical model and performed the optical simulations with B.T.v.G. K.D. and B.T.v.G. fabricated the solar cells. M.M.W. constructed the photocurrent spectrometer. B.T.v.G., M.M.W. and R.A.J.J. planned the research. B.T.v.G. and R.A.J.J. wrote the manuscript, all authors commented on it.

## Competing interests

The authors declare no competing interests.
