## [Peer Review File · Nature Communications]

Revealing defective interfaces in perovskite solar cells from highly sensitive sub-bandgap photocurrent spectroscopy using optical cavitiesREVIEWER COMMENTS

Reviewer #1 (Remarks to the Author):

Referee's Report

Manuscript: "Revealing defective interfaces in perovskite solar cells from highly sensitive sub-bandgap photocurrent spectroscopy using optical cavities"

Authors: van Gorkom et al.

Summary: This work demonstrates the role of interference in the subgap absorption in perovskite materials due to the trap states. The interference patterns (modulated using optical spacers and carefully designed devices) indicate that the deep traps should be at perovskite-fullerene interface rather than the bulk. This is nicely shown in Fig 4. This work also corrects previous publications reporting trap energies in perovskites with total ignorance of optical interference.

General View: Firstly, this is a very well-written manuscript and easy to follow as long as the reader has some basic optics knowledge. The analysis is thorough, and the devices are fabricated very carefully, i.e., the authors have not simply varied the active layer thickness which would result in a change in the crystallinity and nano-morphology. Instead, they employed a sputtered ITO back electrode. To find out where the traps are located (near the fullerene interface) is an important and useful finding and may guide us towards a better trap passivation. It has been very annoying to see some authors have previously taken the subgap EQEs so literally and concluded "multiple trap levels" in perovskites from subgap EQE. This work clearly shows that the position of the peaks is heavily influenced by the cavity interference. This effect has been also shown by Kaiser et al. (ref 31, note year is wrong, the paper is 2019) in organic solar cells where the shape of the trap-dominated subgap EQE is influenced by the cavity interference. Perovskite has large refractive index and perhaps the effect is even more pronounced in this system as shown in this work.

I strongly recommend publication of this study after some minor revision.

I respectfully request authors consider my below comments.

Comments:

- 1- Although Kaiser et al (ref 31) is cited but it might be worth mentioning that they have observed the same effect but in organic solar cells.
- 2- For the reader, it can be obvious that the trap state may absorb sub-gap photons. However, it won't be obvious how they generate charges. The mechanism of

charge generation via these traps can be discussed briefly. The traps should be mid-gap traps (according to SRH statistics) or in other words, if the traps are shallow, their role in SRH recombination is negligible. The mechanism of charge generation via trap states is explained by other researchers (Martin Green for example). This process is '*optical release*' which is the inverse of SRH recombination. This has been explained in ref 21.

- 3- The optical constants of all used layers may be included in the SI.
- 4- In Fig 4, could the authors also consider a scenario where the traps are caused by either the polymer or PCBM layers? (similar to the mid-gap traps seen in organics, ref 31 or 21).
- 5- Would be nice to have representative JV curves in the SI to see that the results are relevant to high efficiency cells.
- 6- These days many researchers are interested in doing ultra-sensitive EQE. It will be nice to explain the sEQE apparatus and how the authors have achieved high sensitivity (nearly -90dB) in the SI. (A similar example is Zeiske et al. ACS Photonics 2020, 7, 256–264).

Ardalan Armin
Swansea University

Reviewer #2 (Remarks to the Author):

This work by van Gorkom et al "Revealing defective interfaces in perovskite solar cells from highly sensitive sub-bandgap photocurrent spectroscopy using optical cavities" is generally a concise, well written manuscript that will be welcomed by the research community. The conclusions are sound and well-supported by the experimental data and the overall narrative is clear and easy to follow. The novelty of the work is clear and the overall concept is insightful and innovative, I recommend that the manuscript be accepted for publication subject to consideration of the following (very) minor points:

- The authors acknowledge challenges associated with modulating the thickness of the perovskite film and the issues with defects in such polycrystalline materials. It is surprising therefore that no structural/morphological characterisation of the devices or perovskite active layers have been provided. This would be useful to allow such properties to be correlated with the high quality optical data. Are the authors able to provide AFM/SEM/TEM/XRD data for the films?

- Please add a caption to Figures 1a-b to help the reader differentiate between the data sets (it is clear in the legend but caption would be useful).

- Some discussion of variations observed with changes in illumination direction (semi-transparent cells) is required as the optical electric field profile will change with such variations.

Reviewer #3 (Remarks to the Author):

This paper reports on an experimental study of the role of defects in perovskite photovoltaic devices using sub-bandgap photocurrent spectroscopy. The main results of the paper are: (i) to point out the effects of optical interference, mainly due to reflections at the aluminum electrode, on the photon energy dependence of the sub-bandgap external quantum efficiency (EQE); (ii) to show that the defects are mainly concentrated at the interface between the perovskite and the electron/hole transporting layers (ETL/HTL), and that, for the particular photovoltaic structure considered in this study, the interface with the PCBM gives the dominant contribution. To prove these results, a series of cavities with spacer layers of different thickness is constructed and their below bandgap photocurrent is modeled using standard transmission transfer methods (TTM). I find both these conclusions not very surprising. In optical systems involving layers with thickness comparable to the light wavelength, as is the case here, optical interference effects are commonly observed, especially when absorption is negligible as in the studied systems. On the other hand, it is well known that defects are mainly generated at the interfaces where dangling bonds are available.

This paper generalizes to perovskite photovoltaic devices the study of the role of cavity effects and optical interferences that have already been described in detail in silicon and organic photovoltaic devices (see e.g. Refs. 30 and 31 of the paper). Also it is not clear if the conclusion that defects are concentrated at the perovskite/PCBM interfaces are generalizable to all perovskite devices or depend on the specific device preparation adopted in this study or on the choice of the ETL. For these reasons, although the paper reports cleanly executed experiments and it is written very clearly, I am not convinced that its conceptual novelty and broad impact are sufficient to justify its publication in Nature Communications. I would rather suggest submission to a more specialized journal.

Response to the Reviewers

We sincerely thank the three Reviewers for their efforts in evaluating our manuscript and very much appreciate their comments and criticism on the contents. Below we have copied the reports of the Reviewers and address the different points raised and outline the changes made in the manuscript.

Reviewer #1

Manuscript: “Revealing defective interfaces in perovskite solar cells from highly sensitive sub-bandgap photocurrent spectroscopy using optical cavities”

Authors: van Gorkom et al.

Summary: This work demonstrates the role of interference in the subgap absorption in perovskite materials due to the trap states. The interference patterns (modulated using optical spacers and carefully designed devices) indicate that the deep traps should be at perovskite-fullerene interface rather than the bulk. This is nicely shown in Fig 4. This work also corrects previous publications reporting trap energies in perovskites with total ignorance of optical interference.

General View: Firstly, this is a very well-written manuscript and easy to follow as long as the reader has some basic optics knowledge. The analysis is thorough, and the devices are fabricated very carefully, i.e., the authors have not simply varied the active layer thickness which would result in a change in the crystallinity and nanomorphology. Instead, they employed a sputtered ITO back electrode. To find out where the traps are located (near the fullerene interface) is an important and useful finding and may guide us towards a better trap passivation. It has been very annoying to see some authors have previously taken the subgap EQEs so literally and concluded “multiple trap levels” in perovskites from subgap EQE. This work clearly shows that the position of the peaks is heavily influenced by the cavity interference. This effect has been also shown by Kaiser et al. (ref 31, note year is wrong, the paper is 2019) in organic solar cells where the shape of the trap-dominated subgap EQE is influenced by the cavity interference. Perovskite has large refractive index and perhaps the effect is even more pronounced in this system as shown in this work.

I strongly recommend publication of this study after some minor revision.

I respectfully request authors consider my below comments.

Reply: Thank you for summarizing the main aspects of our work and the favorable comments. We checked Ref. 31 and although it was indeed first published on line in 2019, it is included in issue 1 of volume 8 of 2020.

Comments:

1- Although Kaiser et al (ref 31) is cited but it might be worth mentioning that they have observed the same effect but in organic solar cells.

Reply: We now mention on page 7 that Kaiser et al. have demonstrated that accounting for interference effects is important when analyzing the sub-bandgap EQE of organic solar cells.

2- For the reader, it can be obvious that the trap state may absorb sub-gap photons. However, it won't be obvious how they generate charges. The mechanism of charge generation via these traps can be discussed briefly. The traps should be mid-gap traps (according to SRH statistics) or in other words, if the traps are shallow, their role in SRH recombination is negligible. The mechanism of charge generation via trap states is explained by other researchers (Martin Green for example). This process is 'optical release' which is the inverse of SRH recombination. This has been explained in ref 21.

Reply: With the method described in the manuscript we have been able to identify the location and the energy of the defects, but it is indeed not obvious how these defects generate charges. The optical release mechanism described in Ref. 21 (and work cited therein) is certainly a viable option. We note that the energies that we identify for the defects of 0.72 and 1.09 eV are not exactly mid-bandgap ($E_g = 1.574$ eV). It might also be that a direct interfacial charge transfer from an occupied trap into the lowest unoccupied molecular orbital of PCBM underlies the observed charge generation below the bandgap. We have added a brief discussion on the mechanism of charge generation via these traps in the Discussion on page 13.

3- The optical constants of all used layers may be included in the SI.

Reply: The refractive indices and extinction coefficients of all used layers have been added to the SI (Supplementary Fig. 3) and this is mentioned on page 4.

4- In Fig 4, could the authors also consider a scenario where the traps are caused by either the polymer or PCBM layers? (similar to the mid-gap traps seen in organics, ref 31 or 21).

Reply: Contributions to the EQE of the charge transport layers have been simulated and are now shown in the SI (Supplementary Fig. 10) and mentioned in the text on page 12.

5- Would be nice to have representative JV curves in the SI to see that the results are relevant to high efficiency cells.

Reply: Representative $J-V$ characteristics and regular EQE spectra for aluminum and ITO top contact cells have been included in the SI (Supplementary Fig. 2) and are referred to on page 4. The opaque cells (Al back electrode) reach PCE of 17.9% and the semitransparent cells (ITO back electrode) reach PCE of 14.7% in reverse scans. In the Methods section (Device characterization.) on page 14 and 15 we detail how the device characterization was performed.

6- These days many researchers are interested in doing ultra-sensitive EQE. It will be nice to explain the sEQE apparatus and how the authors have achieved high sensitivity (nearly -90dB) in the SI. (A similar example is Zeiske et al. ACS Photonics 2020, 7, 256–264).

Reply: The Methods section on the sub-bandgap EQE measurements has been expanded to include more details on the measurement protocol and used settings for the sensitive photocurrent spectroscopy (page 15).

Reviewer #2

This work by van Gorkom et al “Revealing defective interfaces in perovskite solar cells from highly sensitive sub-bandgap photocurrent spectroscopy using optical cavities” is generally a concise, well written manuscript that will be welcomed by the research community. The conclusions are sound and well-supported by the experimental data and the overall narrative is clear and easy to follow.

The novelty of the work is clear and the overall concept is insightful and innovative, I recommend that the manuscript be accepted for publication subject to consideration of the following (very) minor points:

Reply: Thank you for the positive comments on our manuscript.

- The authors acknowledge challenges associated with modulating the thickness of the perovskite film and the issues with defects in such polycrystalline materials. It is surprising therefore that no structural/morphological characterisation of the devices or perovskite active layers have been provided. This would be useful to allow such properties to be correlated with the high quality optical data. Are the authors able to provide AFM/SEM/TEM/XRD data for the films?

Reply: SEM and XRD data have been included in the SI (Supplementary Figure 1). This is mentioned on page 4 of the manuscript. XRD and SEM data are consistent with the typical characteristics of these lead halide perovskites. In the Methods section (Film characterization.) on page 14 we describe the experimental details.

- Please add a caption to Figures 1a-b to help the reader differentiate between the data sets (it is clear in the legend but caption would be useful).

Reply: The nature of the back electrode (Al or ITO) has now been added to Figures 1a-b. This indeed makes it easier to distinguish between the two data sets.

- Some discussion of variations observed with changes in illumination direction (semi-transparent cells) is required as the optical electric field profile will change with such variations.

Reply: The optical electric field ($|E|^2$) was modelled for the two possible illumination directions (glass/ITO and ITO/AZO). Supplementary Fig. S4 shows that $|E|^2$ in the perovskite layer differs only slightly for the different illumination directions. We mention this in the main text on page 5. For the sub-bandgap EQEs that form the basis of the analysis in the manuscript, illumination has always been from the glass/ITO side. Hence the differences $|E|^2$ do not affect any of the conclusions.

Reviewer #3

This paper reports on an experimental study of the role of defects in perovskite photovoltaic devices using sub-bandgap photocurrent spectroscopy. The main results of the paper are: (i) to point out the effects of optical interference, mainly due to reflections at the aluminum electrode, on the photon energy dependence of the sub-bandgap external quantum efficiency (EQE); (ii) to show that the defects are mainly concentrated at the interface between the perovskite and the electron/hole transporting layers (ETL/HTL), and that, for the particular

photovoltaic structure considered in this study, the interface with the PCBM gives the dominant contribution. To prove these results, a series of cavities with spacer layers of different thickness is constructed and their below bandgap photocurrent is modeled using standard transmission transfer methods (TTM). I find both these conclusions not very surprising. In optical systems involving layers with thickness comparable to the lightwavelength, as is the case here, optical interference effects are commonly observed, especially when absorption is negligible as in the studied systems. On the other hand, it is well known that defects are mainly generated at the interfaces where dangling bonds are available. This paper generalizes to perovskite photovoltaic devices the study of the role of cavity effects and optical interferences that have already been described in detail in silicon and organic photovoltaic devices (see e.g. Refs. 30 and 31 of the paper). Also it is not clear if the conclusion that defects are concentrated at the perovskite/PCBM interfaces are generalizable to all perovskite devices or depend on the specific device preparation adopted in this study or on the choice of the ETL. For these reasons, although the paper reports cleanly executed experiments and it is written very clearly, I am not convinced that its conceptual novelty and broad impact are sufficient to justify its publication in Nature Communications. I would rather suggest submission to a more specialized journal.

Reply: Thank you for critically reviewing our work. We agree that the effects of optical interference in sub-bandgap EQE measurements have been noted before for organic and amorphous silicon (a-Si) solar cells. We emphasize this prior work now better on page 8. However, there are numerous papers discussing sub-bandgap measurements on organic and perovskite solar cells without considering interference effects (see also the comment of Reviewer #1). To demonstrate interference on the sub-bandgap photocurrent in organic and a-Si the thickness of the active layer is usually varied, which then causes a change in the sub-bandgap photocurrent. One disadvantage of doing this for perovskites, as is mentioned in the manuscript, is that the quality of the active layer can easily be affected when changing the thickness. In turn this would impact the measured photocurrent because defect densities or optical constants change. The novelty of the optical cavities proposed in the manuscript, is that one can decouple the effects of optical interference and changes in layer quality. The active layers remain untouched and the same device can be measured with different cavity lengths. This has not been shown thus far for any other photovoltaic technology. The method described in this manuscript can therefore not only help understand defect distributions in perovskites devices, but possibly also in other thin film solar cells. In addition, we show that optical interference can be used to identify the regions where the defects are located in the stack without the necessity of making cross sections or removal of layers. Indeed, for perovskites and other (crystalline) semiconductors, undercoordinated ions or dangling bonds are generally accepted to exist predominately at the interfaces and can cause increased defect densities. In that respect it is interesting to note that the results in this manuscript show that for perovskites the defects are located at the top interface and not at the bottom.

REVIEWERS' COMMENTS

Reviewer #1 (Remarks to the Author):

The authors have made appropriate changes and their work is now ready for publication in my opinion. This is a very timely and useful work for perovskite solar cells and photodetector communities.

Ardalan Armin
Department of Physics of Swansea University

Reviewer #2 (Remarks to the Author):

The authors have considered the comments made by all three reviewers and have made appropriate changes to the manuscript. I believe that the paper can be accepted now without further changes.

Reviewer #3 (Remarks to the Author):

In the revised version of the manuscript the authors satisfactorily address the criticism by myself and by the other reviewers and explain more clearly the elements of novelty of this study. In the present form, the paper is in my opinion suitable for publication in Nature Communications.

Response to the Reviewers

Reviewer #1

The authors have made appropriate changes and their work is now ready for publication in my opinion. This is a very timely and useful work for perovskite solar cells and photodetector communities.

Ardalan Armin
Department of Physics of Swansea University

Reply: Thank you for the favorable comments.

Reviewer #2

The authors have considered the comments made by all three reviewers and have made appropriate changes to the manuscript. I believe that the paper can be accepted now without further changes.

Reply: Thank you for the favorable comments.

Reviewer #3

In the revised version of the manuscript the authors satisfactorily address the criticism by myself and by the other reviewers and explain more clearly the elements of novelty of this study. In the present form, the paper is in my opinion suitable for publication in Nature Communications.

Reply: Thank you for the favorable comments.